# Statistical methods applied for the assessment of the HIV cascade and continuum of care: a systematic scoping review

Aneth Vedastus Kalinjuma ,[1,2] Tracy Renée Glass,[3,4] Honorati Masanja,[1] Maja Weisser,[1,3,5] Amina Suleiman Msengwa,[6] Fiona Vanobberghen,[3,4] Kennedy Otwombe [2,7]

FV and KO contributed equally.

For numbered affiliations see end of article.

**Correspondence to**
Dr Aneth Vedastus Kalinjuma; avedastus@ihi.or.tz

## ABSTRACT

**Objectives** This scoping review aims to identify and synthesise existing statistical methods used to assess the progress of HIV treatment programmes in terms of the HIV cascade and continuum of care among people living with HIV (PLHIV).

**Design** Systematic scoping review.

**Data sources** Published articles were retrieved from PubMed, Cumulative Index to Nursing and Allied Health Literature (CINAHL) Complete and Excerpta Medica dataBASE (EMBASE) databases between April and July 2022. We also strategically search using the Google Scholar search engine and reference lists of published articles.

**Eligibility criteria** This scoping review included original English articles that estimated and described the HIV cascade and continuum of care progress in PLHIV. The review considered quantitative articles that evaluated either HIV care cascade progress in terms of the Joint United Nations Programme on HIV and AIDS targets or the dynamics of engagement in HIV care.

**Data extraction and synthesis** The first author and the librarian developed database search queries and screened the retrieved titles and abstracts. Two independent reviewers and the first author extracted data using a standardised data extraction tool. The data analysis was descriptive and the findings are presented in tables and visuals.

**Results** This review included 300 articles. Cross-sectional study design methods were the most commonly used to assess the HIV care cascade (n=279, 93%). In cross-sectional and longitudinal studies, the majority used proportions to describe individuals at each cascade stage (276/279 (99%) and 20/21 (95%), respectively). In longitudinal studies, the time spent in cascade stages, transition probabilities and cumulative incidence functions was estimated. The logistic regression model was common in both cross-sectional (101/279, 36%) and longitudinal studies (7/21, 33%). Of the 21 articles that used a longitudinal design, six articles used multistate models, which included non-parametric, parametric, continuous-time, time-homogeneous and discrete-time multistate Markov models.

**Conclusions** Most literature on the HIV cascade and continuum of care arises from cross-sectional studies.

## STRENGTHS AND LIMITATIONS OF THIS STUDY

⇒ This review is one of the largest and most recent systematic scoping reviews conducted on the methods used to assess the HIV cascade and continuum of care with wide coverage and diversity.

⇒ The review highlights areas with limited research in terms of methods and their applications to the HIV cascade and continuum of care, and it provides recommendations for future research.

⇒ This review was limited to articles published in English. Therefore, the impact of including other languages in this review is unknown.

⇒ The review included published articles. However, the pattern study findings would not be expected to change if grey literature were included.

The use of longitudinal study design methods in the HIV cascade is growing because such methods can provide additional information about transition dynamics along the cascade. Therefore, a methodological guide for applying different types of longitudinal design methods to the HIV continuum of care assessments is warranted.

## INTRODUCTION

The HIV care cascade is a framework that describes the stages of people living with HIV (PLHIV) transitioning through HIV infection, diagnosis, linkage to care, initiation of antiretroviral therapy (ART), retention in care and viral load suppression.[1 2] This framework characterises and quantifies the proportion of PLHIV in each stage of the cascade.[3 4] It is a useful tool for monitoring the uptake of HIV services and the progress of HIV treatment programmes.[5–7]

In the literature, the terms 'HIV care cascade' and 'HIV continuum of care' are used interchangeably,[4 5 8 9] but Kay *et al*[10] distinguished the two concepts. They defined the 'HIV continuum of care' as a dynamic and bidirectional engagement in HIV care,

whereas the 'HIV care cascade' was defined as a static and cross-sectional representation of HIV care stages at the population level. In recent years, many researchers have embraced these concepts, and the distinction between them has been recognised in literature, particularly after the Joint United Nations Programme on HIV and AIDS (UNAIDS) announced the 90-90-90 targets for HIV for 2020[11] and 95-95-95 targets for 2025.[12] The 95-95-95 target states that 95% of PLHIV know their HIV status, 95% of those diagnosed are initiated on ART and 95% of those on ART are virally suppressed.

Currently, most HIV treatment programme assessments and evaluations are performed following the HIV care cascade approach, which aims to report progress towards UNAIDS targets for 2020 or 2025.[13–22] These assessments often apply descriptive statistics and show progress at a particular time point, referred to as a cross-sectional design. In recent years, assessments of the HIV cascade and continuum of care that use a longitudinal design have been increasingly featured in the published literature. In these applications, methods such as multistate models,[3 5] competing risk methods[23] and the distribution time spent in each cascade stage[24] were used. The existence of various methods and their applications in the HIV cascade and continuum of care evaluation call for a review that identifies the statistical methods used in these assessments.

Different types of reviews exist on this topic, the majority of which focus on synthesising estimates for each HIV care cascade stage. These reviews mainly focused on HIV testing, engagement in care and treatment,[25 26] cascades among alcohol users,[27] longitudinal methods for assessing retention in care[28] and cascades among people who inject methadone drug.[29] Others reviewed cascade stage definitions,[30] measures of patient engagement in HIV care,[31] cascade experiences among indigenous people,[32] cascades among adolescents and young adults[33] and cascades among children and adolescents.[34]

Currently, few systematic reviews have synthesised methods used to assess the HIV cascade and continuum of care progress. The first systematic review was conducted by Medland *et al*[35] and published in 2015. At that time, few cascade articles were published, and most studies used descriptive statistics in a cross-sectional study design. Granich *et al*[36] conducted a review of methods and the status of progress towards the UNAIDS 90-90-90 from 2010 to 2016, and they showed that different methods were used. However, they focused mainly on articles that used a cross-sectional study design and methods used to determine indicators and data sources. In 2021, Vourli *et al*[37] published a systematic review that highlighted the importance of expanding the scope of cascade assessment to include time to the next cascade stage. In these reviews, the components of the statistical methods applied to the HIV cascade and continuum of care assessments were limited. Therefore, this systematic scoping review aimed to identify and summarise the statistical methods used to assess HIV treatment programme progress in terms of the

HIV cascade and continuum of care among PLHIV using both cross-sectional and longitudinal study designs.

## METHODS

### Review protocol
The scoping review protocol was developed from November to December 2021, following the Joanna Briggs Institute System for the Unified Management, Assessment, and Review of Information Manual for Evidence Synthesis (Scoping Review Chapter).[38]

### Eligibility criteria
This scoping review included original articles written in English that focused on estimating and describing the HIV cascade and continuum of care among PLHIV. Specifically, the review considered quantitative articles that either evaluated or assessed the HIV care cascade in terms of UNAIDS targets or the dynamics of engagement in HIV care and treatment clinics for the HIV continuum of care. Mixed-methods articles were included if the quantitative component was well described (ie, the method used to assess the cascade and results of the cascade were presented in the article). The review excluded articles comparing the HIV care cascade with other cascades, such as the tuberculosis care cascade, hepatitis cascade and prevention of mother-to-child transmission of HIV. The review further excluded review and meta-analysis articles, opinions, editorials, commentary articles and non-peer-reviewed articles (because the methods sections were not well-described). Finally, the review did not include published conference abstracts or proceedings, because the methods were typically not provided in sufficient detail.

### Search strategy
The development of an electronic bibliographical database search strategy began with a pilot phase, whereby review concepts were searched independently in PubMed and Google Scholar. The retrieved relevant titles and abstracts were screened to identify keywords to include in the development of electronic database search queries. The database search query was first developed in English using the PubMed database and adapted to the Cumulative Index to Nursing and Allied Health Literature (CINAHL) Complete and Excerpta Medica dataBASE (EMBASE) databases as needed (online supplemental table S1). The database search dates were from April to July 2022.

### Source of the evidence selection process
Titles and abstracts were retrieved from all three databases, and duplicates were removed based on the digital object identifier and article content. Titles and abstracts were screened for eligibility by the lead author and irrelevant articles were excluded. All potentially relevant full-text articles were downloaded for review. The first author and two independent reviewers each conducted a full-text

review of all articles, during which the review criteria were applied. The article search and assessment of inclusion and exclusion criteria were summarised and presented using the Preferred Reporting Items for Systematic Reviews and Meta-Analyses extension for Scoping Reviews flow diagram.[39]

## Data extraction and data entry process

The extracted information included the first author's name and publication year, type of participants, age, country, sample size, study design, study level of representation and data source. Other extracted information included cascade stages, cascade size, cascade staging methods and statistical methods used to assess the HIV cascade and continuum of care (online supplemental table S2). All extracted information was first entered into an Excel spreadsheet. The data extraction tool was modified and revised whenever necessary, depending on new information available in the articles and its relevance to this review. The first author reviewed all extracted data. Disagreements between reviewers were resolved by consensus. After finalising the data collection template, an electronic data capture system was developed using open data kit (ODK), and data entry was performed by the reviewers. The information entered into ODK was pooled in Excel with a comma-separated value file format using the ODK briefcase.

## Data items

The variables used in this review included publication year, which included three groups: 2011–2014, referring to the time between the first cascade report and before the announcement of the UNAIDS 90-90-90 targets; 2015–2019, following the UNAIDS 90-90-90 targets announcement and the implementation time and 2020–2022 (by July), representing the reporting time for the UNAIDS 90-90-90 targets and the implementation of the new UNAIDS 95-95-95 targets. The study site was country-specific if a study was conducted in one country and regionally if a study was conducted in at least two countries. The study regions included Asia and Europe, North and South America, sub-Saharan Africa and global (a single report that included multiple regions).

A cross-sectional study design included articles that presented cascade results at a single time point. A longitudinal design included articles that presented cascade outcomes either repeatedly over time (ie, repeated cross-sectional study design), or prospective or retrospective study designs (including cohort data).[40] In this review, studies that included cascade assessments using both cross-sectional and longitudinal designs were included in the longitudinal study design group, because cross-sectional analyses can be considered a subset of longitudinal analyses.

Other variables included data sources (cohort, laboratory test records, hospital records, surveys, surveillance and clinical trial data) and study level of representation, which was grouped into facility, subdistrict/town/

municipality/city; state/province; population (when a study estimated the number of PLHIV, diagnosed and linkage to care); community/district/county; regional; country and global studies. Another variable was the sample size (the number of diagnosed PLHIV assessed). The sample size was grouped into four categories: <1000, 1000–5000, 6000–10 000 and >10 000. The types of participants were grouped into all PLHIV, children only, adolescents, adults, HIV key populations, prisoners, veterans, refugees and the military. The HIV key populations included men who had sex with men, female sex workers, people who inject drugs, and transgender men and women. Finally, the statistical methods applied in the different articles were grouped into two categories: descriptive statistics and regression models. Variables such as study design, data sources, types of participants and statistical methods were allowed for multiple responses; the obtained responses were converted into multiple individual variables with binary responses.

Stage names with similar meanings were reorganised to formulate simple cascade stage names, such as HIV infection (PLHIV), diagnosis, linkage to care, loss to follow-up (LTFU), retention in care, transfer to other HIV clinics, ART eligibility, on ART, viral load suppression and death. Articles that used a longitudinal design and multistate models had slightly different names for their cascade stages. In this case, stage names were used as they were written by the authors. The size of the cascade and continuum of care frameworks was categorised into four groups based on the number of stages assessed. The four categories were extralarge cascades (>six stages), large cascades (five to six stages), medium cascades (four stages) and small cascades (≤three stages).

The cascade frameworks used were grouped into six: the Centers for Disease Control and Prevention (CDC) 2011 cascade framework (including PLHIV, diagnosis, linkage to care, retention in care, ART status and viral load suppression status stages[2]). CDC with five stages (starting from the diagnosis, linkage to care, retention in care, ART status and viral load suppression status stage), and CDC with four stages (starting from linkage to care, retention in care, ART status and viral load suppression status stages). Gourlay and others in 2017 (here abbreviated as Gourlay 2017) included PLHIV, diagnosis, ART status and viral load suppression stages.[41] Gourlay 2017 with three stages started from the diagnosis. Other cascade formulations included stages such as ART eligibility, clinical staging, ART adherence, LTFU/disengagement, engagement in care, death, transfer out, CD4 testing, current use of ART and history of ART. The cascade staging method used was grouped into three categories: dependent (also known as conditional), independent and a combination of the two. In the dependent method, PLHIV are eligible for the next cascade stage if they are in the previous cascade stage, whereas in the independent method, each cascade stage was derived independently of the prior cascade stage.[42]

## Data analysis and presentation

The review findings were analysed and summarised using summary statistics, such as counts with percentages, medians with IQRs, and data ranges. Results were presented using tables. The geographical distribution of articles by country was summarised using world maps. Maps were plotted for country-specific articles (excluding 28 regional articles). The constructed cascade and continuum of care frameworks were summarised in terms of the size and cascade stages used. Types of cascade framework used were summarised using counts with percentage. The frameworks of a subset of articles that used a longitudinal study design and multistate models were compiled to document similarities, differences and existing gaps. All data management and statistical analyses were performed using Stata software V.14 (StataCorp LP, College Station, Texas, USA).

## Patient and public involvement

Patients and/or the public were not involved.

## RESULTS
### Description of the article selection process

The search query retrieved 5805 titles and abstracts from all three databases, 5495 titles were excluded in the title and abstract screening stages (figure 1). The 310 abstracts were identified as meriting full-text reviews, of which 300 articles were included (online supplemental table S3).

### Characteristics of articles

The majority of the 300 articles were published between 2015 and 2019 (n=177, 63%) (table 1). Various PLHIV groups were assessed, including children, adolescents, adults, migrants, prisoners, veterans, the military and HIV key populations. HIV key populations were reported in 74 articles, which included men having sex with men (n=35, 47%), female sex workers (n=19, 26%), people who inject drugs (n=20, 27%), and transgender men (n=8, 11%) and women (n=19, 26%).

Assessments of the HIV cascade and continuum of care were performed using cross-sectional (279/300, 93%) and longitudinal (21/300, 7%) designs (table 1). The review included 28 studies conducted in regions or globally. Many of these studies were conducted in Asia and Europe (n=11, 39%) and in sub-Saharan Africa (n=11, 39%). Focusing on country-specific studies, most of the articles that used a cross-sectional study design were from the USA, Canada, Zimbabwe and South Africa (these countries each contributed between 9 and 83 of 252 studies) (figure 2). Articles that used a longitudinal study design were mainly from the USA (6/20 studies).

Various data sources have been used to describe the HIV cascade and continuum of care. In the 279 articles that used a cross-sectional study design, the survey data source was common (n=110, 39%), whereas, in the 21 articles that used a longitudinal design, the cohort data source was often used (n=10, 48%) (table 1). The sample size varied across articles, with the smallest study including 22 hard-to-reach individuals and the largest study including 37.9 million PLHIV (UNAIDS report of 170 countries). In articles that used a cross-sectional study design, the median sample size was 1721 PLHIV (IQR: 524–8744), whereas in articles that used a longitudinal design, the median was 3311 PLHIV (IQR: 1080–11510). Articles that used a cross-sectional study design had various levels of representation, with many being subdistrict/town/municipality/city-based studies (68/278, 25%), while for articles that used a longitudinal design, the majority were facility-based studies (6/18, 33%).

### HIV cascade and continuum of care stages

In general, a median of four cascade stages (IQR: 3–5) was used (table 2). The smallest cascade had two stages, and the largest cascade had nine stages. Both Cross-sectional studies and longitudinal studies used large cascades (five to six stages) (95/279, 34% and 7/21, 33% respectively) and small cascades (≤three stages) (94/279, 34% and 7/21, 33% respectively). In articles that used a longitudinal design, the larger the cascade size the larger the median sample size. However, there was no clear pattern between sample size and the number of stages used in articles that used cross-sectional study design methods (online supplemental figure S1).

In articles that used a cross-sectional study design, 12% (34/279) used Gourlay 2017 cascade framework (table 2). The CDC 2011 cascade framework was used in various forms: 15/279 (5%) studies used all six stages, and others used the shorter form with either five stages (9/279, 3%) or four stages (11/279, 4%). In 21 articles that used a longitudinal study design, few used CDC 2011 framework (n=1, 5%) and Gourlay 2017 (n=1, 5%). In all articles reviewed, 71% (190/267) used dependent staging methods. In articles that used a cross-sectional study design, only 7% (17/255) used both dependent and independent methods (table 2).

In both study designs, common stages were PLHIV, diagnosis, linkage to care, ART status, viral load suppression, death and LTFU (online supplemental table S4). In articles that used a longitudinal design, some used other stages such as optimal care, suboptimal care, disengaged and no ART. Lastly, two articles used a longitudinal design with formulated stages in combination with either ART status or CD4 cell count (online supplemental figure S2).

### Statistical methods applied

In the descriptive analyses, the majority of articles used proportions to describe participants at each cascade stage (cross-sectional design (276/279, 99%) and longitudinal design (20/21, 95%)) (table 3). Articles that used a longitudinal study design often estimated transition probabilities between cascade stages (5/21, 24%), which was rare among studies that used a cross-sectional design (1/279, 0.4%). Furthermore, the time spent in each cascade stage (5/21, 24%) and the Kaplan-Meier or cumulative incidence function (5/21, 24%) were often estimated in the

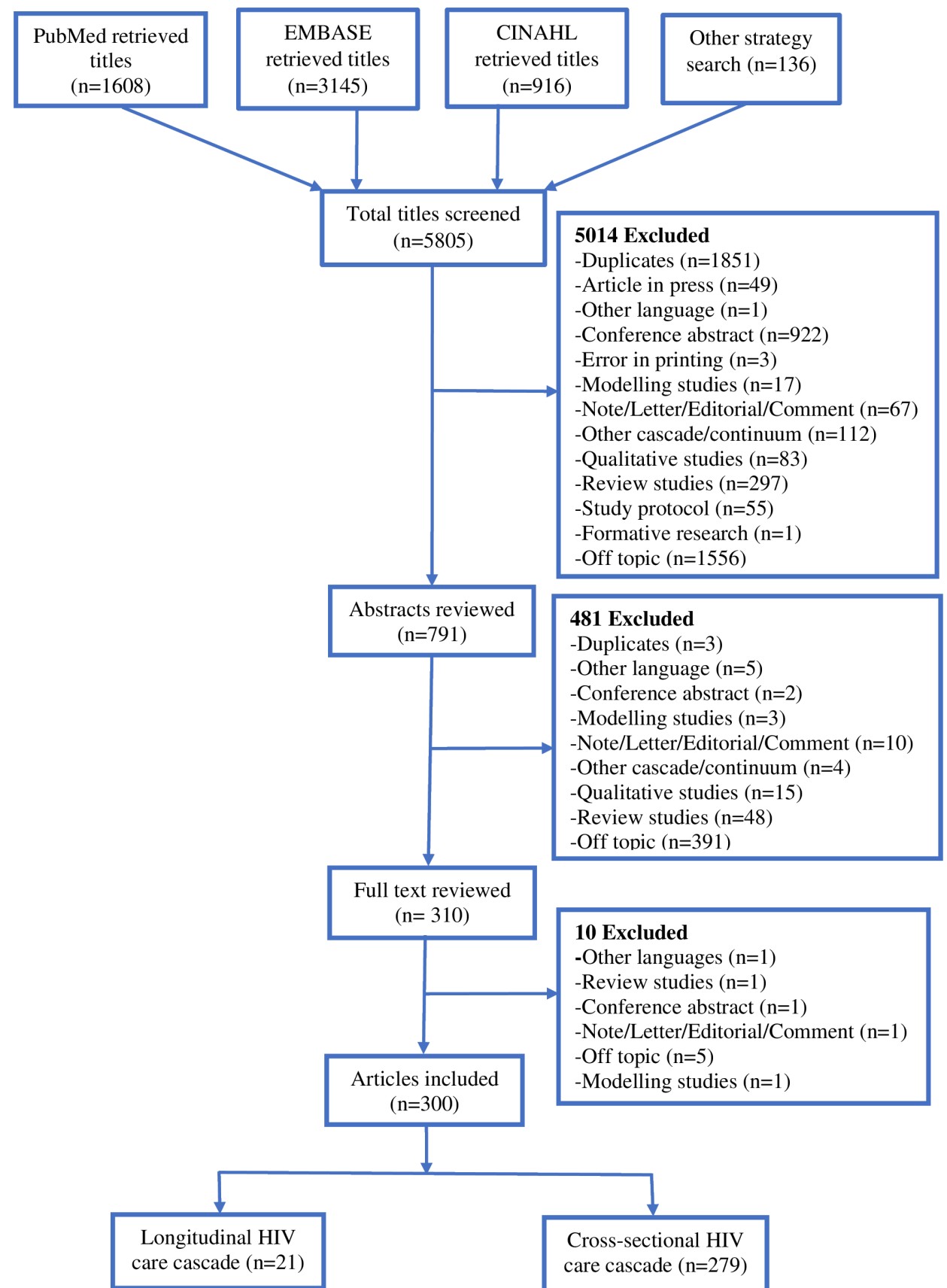

**Figure 1** Flow chart of the inclusion and exclusion of research articles. CINAHL, Cumulative Index to Nursing and Allied Health Literature; EMBASE, Excerpta Medica dataBASE.

**Table 1** Characteristics of articles included in the scoping review (n=300)

| Characteristics of articles | Cross-sectional | Longitudinal* | Overall |
|---|---|---|---|
| Papers included† | 279/300 (93.0%) | 21/300 (7.0%) | 300/300 (100%) |
| **Publication year** | | | |
| 2011–2014 (before 90-90-90)‡ | 18/279 (6.5%) | 0/21 (0%) | 18/300 (6.0%) |
| 2015–2019 (post 90-90-90) | 177/279 (63.4%) | 14/21 (66.7%) | 191/300 (63.7%) |
| 2020–2022 (reporting of both)§ | 84/279 (30.1%) | 7/21 (33.3%) | 91/300 (30.3%) |
| **Regional studies (involved ≥two countries)** | | | |
| Asia and Europe | 10/27 (37.4%) | 1/1 (100%) | 11/28 (39.3%) |
| North and South America | 5/27 (18.5%) | 0/1 (0%) | 5/28 (17.9%) |
| Sub-Saharan Africa | 11/27 (40.7%) | 0/1 (0%) | 11/28 (39.3 %) |
| Global¶ | 1/27 (3.7%) | 0/1 (0%) | 1/28 (3.6%) |
| **Type of participants**** | | | |
| Mixture (all PLHIV) | 47/279 (16.9%) | 3/21 (14.3%) | 50/300 (16.7%) |
| Children only | 10/279 (3.6%) | 1/21 (4.8%) | 11/300 (3.7%) |
| Adolescents | 29/279 (10.4%) | 5/21 (23.8%) | 34/300 (11.3%) |
| Adult | 127/279 (45.5%) | 14/21 (66.7%) | 141/300 (47.0%) |
| Key population | 72/279 (25.8%) | 2/21 (9.5%) | 74/300 (24.7%) |
| Migrants | 5/279 (1.8%) | 0/21 (0%) | 5/300 (1.7%) |
| Prisoners | 2/279 (0.7%) | 0/21 (0%) | 2/300 (0.7%) |
| Veterans | 2/279 (0.7%) | 0/21 (0%) | 2/300 (0.7%) |
| Refugees | 0/279 (0%) | 1/21 (4.8%) | 1/300 (0.3%) |
| Military | 0/279 (0%) | 1/21 (4.8%) | 1/300 (0.3%) |
| **Types of HIV key population**,††** | | | |
| Men having sex with men | 33/72 (45.8%) | 2/2 (100%) | 35/74 (47.3%) |
| Female sex workers | 19/72 (26.4% | 0/2 (0%) | 19/74 (25.7%) |
| People who inject drugs | 20/72 (27.8%) | 0/2 (0%) | 20/74 (27.0%) |
| Transgender women | 19/72 (26.4%) | 0/2 (0%) | 19/74 (25.7%) |
| Transgender men | 7/72 (9.7%) | 1/2 (50.0%) | 8/74 (10.8%) |
| **Data sources**** | | | |
| Cohort data | 65/279 (23.3%) | 10/21 (47.6%) | 75/300 (25.0%) |
| Laboratory test records | 11/279 (3.9%) | 0/21 (0%) | 11/300 (3.7%) |
| Hospital medical records | 55/279 (19.7%) | 5/21 (23.8%) | 60/300 (20.0%) |
| Survey data | 110/279 (39.4%) | 3/21 (14.3%) | 113/300 (37.7%) |
| Surveillance data | 70/279 (25.1%) | 3/21 (14.3%) | 73/300 (24.3%) |
| Clinical trials data | 4/279 (1.4%) | 1/21 (4.8%) | 5/300 (1.7%) |
| Others‡‡ | 19/279 (6.8%) | 3/21 (14.3%) | 22/300 (7.3%) |
| Sample size, median (IQR) | 1721 (524–8744) | 3311 (1080–11510) | 1862 (545–8859) |
| Sample size range (minimum, maximum) | 22–39 000 000 | 100–92 215 | 22–39 000 000 |
| **Sample size (number of participants)** | | | |
| <1000 | 103/279 (37.3%) | 5/21 (21.8%) | 108/297 (36.4%) |
| 1000–5000 | 85/279 (30.8%) | 7/21 (33.3%) | 92/297 (31.0%) |
| 6000–10 000 | 24/279 (8.7%) | 3/21 (14.3%) | 27/297 (9.1%) |
| >10 000 | 64/279 (23.2%) | 6/21 (28.6%) | 70/297 (23.6%) |
| **Level of representation** | | | |
| Facility§§ | 51/278 (18.4%) | 6/18 (33.3%) | 57/296 (19.3%) |
| Subdistrict/town/municipality/city | 68/278 (24.5%) | 2/18 (11.1%) | 70/296 (23.7%) |

**Table 1** Continued

| Characteristics of articles | Cross-sectional | Longitudinal* | Overall |
|---|---|---|---|
| State/province | 36/278 (13.0%) | 0/18 (0%) | 36/296 (12.2%) |
| Population¶¶ | 52/278 (18.7%) | 2/18 (11.1%) | 54/296 (18.2%) |
| Community/district/county | 31/278 (11.2%) | 3/18 (16.7%) | 34/296 (11.5%) |
| Regional | 17/278 (6.1%) | 2/18 (11.1%) | 19/296 (6.4%) |
| Country | 22/278 (7.9%) | 3/18 (16.7%) | 25/296 (8.5%) |
| Global¶ | 1/278 (0.4%) | 0/18 (0%) | 1/296 (0.3%) |

*Longitudinal design articles included articles that conducted both longitudinal and cross-sectional analyses.
†Row percentage.
‡The UNAIDS 90-90-90 target of the year 2020 stated that 90% of PLHIV know their HIV status, 90% of the diagnosed positive are initiated on ART and 90% of those on ART are virally suppressed.
§Articles published in 2020-2022 assessed both UNAIDS targets for 2020 (90-90-90) and 2025 (95-95-95).
¶Global studies are the studies included multiple regions such as Europe, Asia, America and sub-Saharan Africa.
**Percentage was estimated for each type of response and the report in this table is for 'yes' for all binary response variables processed from multiple responses variables.
††Analysis was restricted among the articles reported in the key population participants (n=74).
‡‡Other data sources largely included clinic and programme registries, databases, different types of medical records, consortiums, population estimates reports, social insurance schemes and AIDS programme databases.
§§Ficility-level studies at a health facility or clinic or hospital-based studies.
¶¶Population studies are the studies that included population-level cascade stages which included PLHIV, diagnosis and linkage to care.
ART, antiretroviral therapy; IQR, interquartile range; PLHIV, people living with HIV; UNAIDS, Joint United Nations Programme on HIV and AIDS.

longitudinal studies. Approximately 28% (84/300) of the included articles used simple test statistics, such as the $\chi^2$ test, Fisher's exact test, t-test, Wilcoxon rank-sum test and Kruskal-Wallis test.

To assess the association between cascade stages or transitions and participant characteristics, 176/300 (59%) of the included articles performed regression analysis. The regression models used in the reviewed articles primarily

**A** The geographical distribution of cross-sectional individual studies

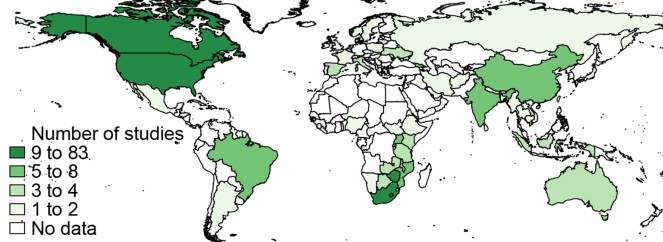

**B** The geographical distribution of longitudinal individual studies

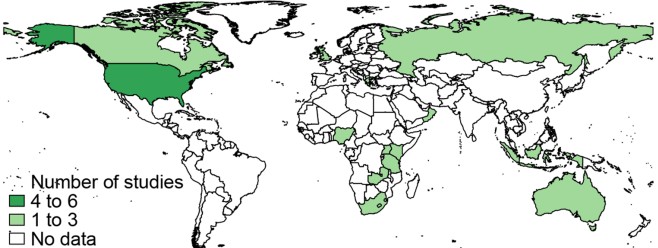

**Figure 2** The geographical location distribution of country-specific studies of cross-sectional (n=252) and longitudinal (n=20) studies (regional studies, such as sub-Saharan Africa or Asia and Europe, were excluded).

considered discrete and survival outcomes (table 3). In the articles that used a cross-sectional study design, the logistic regression model was common (101/279, 36%). In studies that used a longitudinal design, logistic and Cox proportional hazard models were commonly used (33% (7/21) and 19% (4/21), respectively).

### Longitudinal design with multistate models
Among the 21 articles that used a longitudinal study design, different analysis methods were used. Articles that used longitudinal analysis methods were mainly in five groups (online supplemental table S5): classic survival analyses (n=2), survival analysis with competing risk methods (n=5), repeated cross-sectional and time spent in each cascade method (n=2), survival analysis with multistate models (n=6) and repeated measure analysis methods (n=1).

In articles that used a longitudinal study design and multistate models (n=6), five types of multistate models were used. These included non-parametric multistate; parametric, continuous-time multistate; time-homogeneous Markov models; discrete-time multistate Markov and general multistate models (table 4). Three articles adopted the Markov process, and one study considered the Markov process with first-order dependence. In almost all reviewed articles (n=5), the assessment of the care cascade started at enrolment in HIV care, and one article considered ART initiation as the baseline. Regardless of the type of multistate model used, death was considered an absorbing state in all the reviewed articles. Other absorbing states included LTFU

**Table 2**  Summary of various forms of outcomes used in the articles included in the scoping review (n=300)

| Outcome | Cross-sectional | Longitudinal* | Overall |
|---|---|---|---|
| Number of cascade stages, median (IQR) | 4 (3–5) | 4 (3–5) | 4 (3–5) |
| Size of the HIV care cascade | | | |
| Extralarge cascade (>six stages) | 15/279 (5.4%) | 1/21 (4.8%) | 16/300 (5.3%) |
| Large cascade (five to six stages) | 95/279 (34.1%) | 7/21 (33.3%) | 102/300 (34.0%) |
| Medium cascade (four stages) | 75/279 (26.8%) | 6/21 (28.6%) | 81/300 (27.0%) |
| Small cascade (≤three stages) | 94/279 (33.7%) | 7/21 (33.3%) | 101/300 (33.7%) |
| Sample size by size of the HIV care cascade, median (IQR) | | | |
| Extralarge cascade (>six stages) | 3295 (595–7489) | NA† | 3674 (645–10315) |
| Large cascade (five to six stages) | 1418 (500–7843) | 7810 (1532–32242) | 1532 (592–8382) |
| Medium cascade (four stages) | 2355 (440–10076) | 3287 (2196–6850) | 2617 (463–9829) |
| Small cascade (≤three stages) | 1999 (579–9150) | 476 (175–11510) | 1931 (528–9150) |
| Types of cascade framework used | | | |
| CDC 2011‡ | 15/279 (5.4%) | 1/21 (4.8%) | 16/300 (5.3%) |
| Gourlay 2017§ | 34/279 (12.2%) | 1/21 (4.8%) | 35/300 (11.7%) |
| CDC 2011 with five stages¶ | 9/279 (3.2%) | 0/21 (0%) | 9/300 (3.0%) |
| CDC 2011 with four stages** | 11/279 (3.9%) | 0/21 (0%) | 11/300 (3.8%) |
| Gourlay 2017 with three stages†† | 12/279 (4.3%) | 0/21 (0%) | 12/300 (4.0%) |
| Other‡‡ | 198/279 (71.0%) | 19/21 (90.5%) | 217/300 (72.3%) |
| Cascade staging approaches§§ | | | |
| Dependent¶¶ | 183/255 (71.8%) | 7/12 (58.3%) | 190/267 (71.2%) |
| Independent*** | 55/255 (21.6%) | 4/12 (33.3%) | 59/267 (22.1%) |
| Both dependent and independent | 17/255 (6.9%) | 1/12 (8.3%) | 18/267 (6.7%) |

*Longitudinal design articles included articles that conducted both longitudinal and cross-sectional analyses.
†NA is not applicable because there was one study under this group with a sample size of 23 227.
‡Six cascade stages were used as follows: PLHIV, diagnosis, linkage in care, retention in care, on ART and viral load suppression.
§Four stages were used as follows: PLHIV, diagnosis, on ART and viral load suppression.
¶The cascade adopted stages like CDC 2011, but the assessment started from the diagnosis stage
**The cascade adopted stages like CDC 2011, but the assessment started from linkage to care stage.
††The cascade adopted stages like Gourlay 2017, but the assessments started from the diagnosis.
‡‡Other cascade formulations that included other stages and longitudinal frameworks.
§§Studies used longitudinal and multistate models or survival analyses; staging methods for each cascade stage are not applicable.
¶¶In the dependent method, PLHIV are eligible for the next cascade stage if they achieved the prior cascade stage.
***In the independent method, each cascade stage is derived independently of the prior cascade stage.
ART, antiretroviral therapy; CDC, Centers for Disease Control and Prevention; IQR, interquartile range; PLHIV, people living with HIV.

and transfer to other HIV clinics. Three studies used a model that allowed back and forth for LTFU and disengagement states (online supplemental figure S2).

## DISCUSSION

The findings of this scoping review showed that cross-sectional study design methods were most commonly used to assess the progress of HIV service uptake and treatment responses (93%). The reviewed articles were mainly available at the subnational level. The articles used a cross-sectional study design covering a wide range of PLHIV, including children, adolescents, adults, migrants,

prisoners and key populations. In general, the cascade size had a median of four stages (IQR: 3–5). The dependent staging method was often used in articles with a cross-sectional study design. Various longitudinal design methods were used, including repeated cross-sectional, repeated measures and survival analyses. In articles that used a longitudinal study design and multistate methods (n=6 out of 21 articles), only three articles used analytical frameworks that integrated the in- and out-of-care movements of PLHIV enrolled in care. Although cascade researchers pointed out challenges of using a cross-sectional design in 2015,[4 6] articles using a longitudinal

**Table 3** Statistical methods used in articles included in the scoping review (n=300)

| Methods | Cross-sectional | Longitudinal* | Overall |
|---|---|---|---|
| Descriptive statistics† | | | |
| Proportion for each stage of the HIV care cascade | 276/279 (98.9%) | 20/21 (95.2%) | 296/300 (98.7%) |
| Simple test statistics‡ | 78/279 (28.0%) | 6/21 (28.6%) | 84/300 (28.0%) |
| Time spent in each stage of the HIV care cascade | 7/279 (2.5%) | 5/21 (23.8%) | 12/300 (4.0%) |
| Transition probabilities§ | 1/279 (0.4%) | 5/21 (23.8%) | 6/300 (2.0%) |
| Kaplan-Meier or cumulative incidence function | 0/279 (0%) | 5/21 (23.8%) | 5/300 (1.7%) |
| Regression models† | | | |
| Logistic model | 101/279 (36.2%) | 7/21 (33.3%) | 108/300 (36.0%) |
| Cox Proportional Hazard model | 15/279 (5.4%) | 4/21 (19.1%) | 19/300 (6.3%) |
| Poisson model | 32/279 (11.5%) | 0/21 (0%) | 32/300 (10.7%) |
| Competing risk model | 3/279 (1.1%) | 0/21 (0%) | 3/300 (1.0%) |
| Generalised linear model | 10/279 (3.5%) | 0/21 (0%) | 10/300 (3.3%) |
| Weighted and unweighted linear model | 4/279 (1.4%) | 0/21 (0%) | 4/300 (1.3%) |
| Log-binomial regression model | 5/279 (1.8%) | 0/21 (0%) | 5/300 (1.7%) |
| Multinomial regression model | 0/279 (0%) | 3/21 (14.3%) | 3/300 (1.0%) |
| Generalised estimating equation model | 1/279 (0.4%) | 0/21 (0%) | 1/300 (0.3%) |
| Other models¶ | 19/279 (6.8%) | 2/21 (9.5%) | 21/300 (7.0%) |

*Longitudinal design articles included articles that conducted both longitudinal and cross-sectional analyses.
†Percentage was estimated for each type of response and the report in this table is for yes for all binary response variables processed from multiple responses variables
‡Simple test statistics used included the $\chi^2$ test, Fisher's exact test, t-test, Wilcoxon rank -sum test and Kruskal-Wallis test.
§Transition probability is a predicted probability of moving between two consecutive states in a given time.
¶Other models applied were ARIMA or SARIMA, Bayesian hierarchical models, bivariable and multivariable regression, generalised additive model, negative binomial model, generalised linear mixed model and spectrum model.
ARIMA, Autoregressive Intergrated Moving Average model; PH, Proportional Hazard; SARIMA, Seasonal Autoregressive Intergrated Moving Average model.

design and addressing the in- and out-of-care pathways in cascade assessments are scarce. The major reason being these methods require advanced data management and analysis skills.[43 44]

Several authors have proposed different types of frameworks that include various pathways through the HIV cascade and continuum of care. The 'HIV States and Transition' framework was formulated to capture the interaction of PLHIV with HIV testing, care and treatment services at a given time.[6] It has two key features: mutually exclusive states (cascade stages) and transitions between states. This framework was adapted by the articles that expanded the cascade assessment using longitudinal design with multistate methods (online supplemental figure S2).[5 8 45–47] Others proposed the 'Comprehensive HIV Care Cascade', which is a framework formulated by combining stages used in the standard HIV care cascade with the additional consideration of poor outcomes (LTFU or death before or after ART initiation).[48] The 'Cyclical Cascade of HIV Care' was proposed to capture the non-linearity of PLHIV engagement in HIV treatment clinics.[49] The framework also maintained stages of the standard HIV care cascade (diagnosis, linkage and ART initiation) with additional stages of early retention

(<6 months) and long retention (>6 months). Disengagement was incorporated with the possibility of re-engaging in care.

In this review, less than half (n=83, 28%) used either the CDC 2011 or Gourlay 2017 cascade frameworks to quantify and describe the HIV cascade and continuum of care. Other researchers modified the cascade for various reasons. First, data availability, which affects mostly stages included; for example, some studies started with PLHIV linked in care,[50–52] while others did not have viral load data.[53] In articles that used longitudinal design and multistate methods, the lack of transfer-out information[8] and failure to capture re-engagement in care after LTFU[46] affected the inclusion of these stages. Second, the HIV management guidelines used during the cascade assessments changed over time. This resulted in the inclusion of cascade stages such as ART eligibility[54] and CD4 testing.[55] Third, additional information on the cascade. This was the case when ART status was further divided into two stages ever on ART or history of ART use and current use of ART.[56–58] Others added the ART adherence stage[56 59 60] or used the retention stage in both pre-ART and post-ART initiation.[61] Lastly, addressing the in- and out-of-care transitions. In assessments that used a longitudinal design with

**Table 4** Articles that used longitudinal study design and multistate model (n=6)

| Methods | Number of articles |
|---|---|
| Baseline* | |
| Enrolment/recruitment | 5 |
| ART initiation | 1 |
| Reason for censoring† | |
| End of follow-up | 2 |
| Transfer, death, and follow-up | 1 |
| Transfer out, death, and status less than 12 months | 1 |
| Absorbing‡ state | |
| Death | 2 |
| Death and LTFU | 2 |
| Death and transfer out | 2 |
| Types of multistate models | |
| Non-parametric§ multistate model | 1 |
| Parametric¶, continuous-time** multistate | 1 |
| Time-homogeneous†† Markov model | 1 |
| Discrete-time‡‡ multistate Markov model | 2 |
| General multistate model | 1 |
| Multistate model assumptions | |
| First-order dependence | 1 |
| Markov process§§ | 2 |
| Markov, first-order dependence | 1 |
| Missing data techniques | |
| Multiple imputations | 2 |
| Missing data indicator | 3 |

*Baseline is the time at which the assessment of the cascade started.
†Censoring occurs when the exact survival time is unknown from the start until the end of the study.
‡Absorbing state is a state in which once visited no out-of-state transition is possible.
§Non-parametric model is the model that does not take any distribution assumption.
¶Parametric model is the model that follows a certain distribution, for example, exponential and Weibull.
**Continuous-time model is the model used in a dataset collected in unequal-spaced time.
††Time-homogenous model is the model that assumes transition intensity rates are constant over time.
‡‡Descrite-time model is the model used in a dataset collected in the equally spaced time interval.
§§Markov assumption is the assumption used in multistate models that assumes the future transition is only dependent on the current state.
ART, antiretroviral therapy; LTFU, loss to follow-up.

multistate methods, engagement in care was expanded to capture PLHIV who were LTFU or disengaged from care and later returned to care.[5 45]

Almost all articles quantified the cascade using the proportion of PLHIV achieving each cascade stage.

Quantification was done either by using a dependent or independent staging method.[17] The independent staging method is often used in combination with the dependent method, but it is used mainly to assess the proportion of PLHIV with suppressed viral load.[62] This is because the independent staging method reflects the actual uptake of health services,[63] while dependent staging methods mainly focus on PLHIV retained in care.[16 64] Furthermore, approximately 60% of the articles performed regression analyses to examine the relationship between patient characteristics and either each of the cascade stages[54 65–68] or a few selected cascade stages.[69–73] In these assessments, modifiable characteristics were identified to plan for interventions that will improve cascade progress in stages with poor progress.

This review revealed different longitudinal design methods. Assessments were mostly conducted using survival methods. These were mainly in three groups: classic survival analysis, competing risk methods and multistate models. Competing risk methods were used primarily used to estimate the cumulative probabilities function of each stage,[17 23] whereas, in multistate methods, the aim was to capture all possible transitions between stages.[5 47]

The multistate methods used included parametric and non-parametric multistate, continuous-time and discrete-time multistate, and general multistate models. A multistate model is a continuous-time stochastic process that allows individuals to move between a finite number of states.[74] In the HIV cascade and continuum of care, states correspond to the HIV cascade stages.[5 6 45] In multistate modelling frameworks, states can be either transient (participants can move in and out of the state) or absorbing (no transition is possible out of the state).[75] In this review, articles that used multistate models considered death and transfer to other HIV clinics as absorbing states. Depending on the availability of data, LTFU and disengagement from care were allowed to be transient or absorbing states (online supplemental figure S2).

This review is among the largest and most recent systematic scoping reviews conducted on the methods used to assess the HIV cascade and continuum of care with extensive coverage and diversity. Furthermore, because of its coverage, the review identifies areas with limited research in terms of methods and their applications to the HIV cascade and continuum of care and provides recommendations for future studies. However, there were some limitations. First, this review was restricted to English articles. The impact of including other languages in this review is unknown, but areas with a high burden of HIV are mostly anglophone countries. Therefore, the pattern of review findings is not expected to change. Second, the review included peer-reviewed articles. The impact of omitting grey literature is also unknown, but because cross-sectional design methods are widely used, the conclusion of this review is unlikely to be affected by the missed grey literature.

## CONCLUSION

Most studies on the HIV cascade and continuum of care applied cross-sectional study design methods. The use of longitudinal study design methods in the assessment of the HIV cascade is increasing. These methods provide additional information about the transition dynamics along the cascade, establish detailed progress and identify additional areas for interventions. Multistate modelling frameworks can be viewed as extensions of established cross-sectional design frameworks regarding engagement in HIV care, but further work is needed to integrate ART status and viral load suppression status into these frameworks. Therefore, a methodological guide on the application of different types of longitudinal design methods in the HIV cascade and continuum care and their corresponding frameworks is needed to guide future studies.

**Author affiliations**
[1]Department of Interventions and Clinical Trials, Ifakara Health Institute, Ifakara, Dar es Salaam, United Republic of Tanzania
[2]School of Public Health, Faculty of Health Sciences, University of the Witwatersrand, Johannesburg, Gauteng, South Africa
[3]Medicines Department, Swiss Tropical and Public Health Institute, Basel, Switzerland
[4]University of Basel, Basel, Switzerland
[5]Division of Infectious Diseases and Hospital Epidemiology, University Hospital Basel, University of Basel, Basel, Switzerland
[6]Department of Statistics, University of Dar es Salaam, Dar es Salaam, United Republic of Tanzania
[7]Perinatal HIV Research Unit, Chris Hani Baragwanath Academic Hospital, Faculty of Health Sciences, University of the Witwatersrand, Johannesburg, South Africa

**Acknowledgements** Apatsa Selemani: Librarian and CARTA fellow at the Kamuzu University of Health Sciences, Malawi (formerly College of Medicine and Kamuzu College of Nursing of the University of Malawi), Malawi. The librarian contributed to the development of the review search strategies for various databases. Samwel Godfrey Lwambura (Statistician and Data Officer) and Jacqueline Minja (Statistician and Research Scientist), Ifakara Health Institute, Tanzania, research assistants, assisted with full-text review, data extraction, and data entry. Dr Yeromin Mlacha (Medical Entomologist and Senior Researcher), Ifakara Health Institute, Tanzania, assisted in the data preparation for maps of the included articles.

**Contributors** AVK designed the study, obtained funding, conducted the review and prepared the manuscript. KO and FV supervised the study design and implementation. TRG, MW, ASM and HM participated in the study design and reviewed the manuscript. All authors have reviewed and approved the final manuscript. AVK takes the overall repsonsibility of the contents as guarantor.

**Funding** This research was supported by the Consortium for Advanced Research Training in Africa (CARTA). CARTA is jointly led by the African Population and Health Research Center and the University of the Witwatersrand, and funded by the Carnegie Corporation of New York (Grant No. G-19-57145), Sida (Grant No. 54100113), Uppsala Monitoring Center, Norwegian Agency for Development Cooperation (Norad), and by the Wellcome Trust (reference no. 107768/Z/15/Z) and the UK Foreign, Commonwealth & Development Office, with support from the Developing Excellence in Leadership, Training and Science in Africa (DELTAS Africa) programme. The statements made and views expressed are solely the responsibility of the Fellow.

**Map disclaimer** The inclusion of any map (including the depiction of any boundaries therein), or of any geographic or locational reference, does not imply the expression of any opinion whatsoever on the part of BMJ concerning the legal status of any country, territory, jurisdiction or area or of its authorities. Any such expression remains solely that of the relevant source and is not endorsed by BMJ. Maps are provided without any warranty of any kind, either express or implied.

**Competing interests** 'Yes, there are competing interests for one or more authors and I have provided a Competing Interests statement in my manuscript and in the box below'

**Patient and public involvement** Patients and/or the public were not involved in the design, conduct, reporting or dissemination plans of this research.

**Patient consent for publication** Not applicable.

**Ethics approval** This study is part of the PhD project that was approved by the University of the Witwatersrand Human Research Ethics Committee (Medical) with reference number R14/49, however, this study did not involve human participants.

**Provenance and peer review** Not commissioned; externally peer reviewed.

**Data availability statement** Data are available upon reasonable request. All review data extracted and processed is available upon request through the corresponding author.

**ORCID iDs**
Aneth Vedastus Kalinjuma http://orcid.org/0000-0001-5862-9264
Kennedy Otwombe http://orcid.org/0000-0002-7433-4383

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
