## [Reviewer comments · BMJ Open]

ARTICLE DETAILS

TITLE (PROVISIONAL)	Statistical Methods Applied for the Assessment of the HIV Cascade and Continuum of Care: A Systematic Scoping Review
AUTHORS	Kalinjuma, Aneth Vedastus; Glass, Tracy Renée; Masanja, Honorati; Weisser, Maja; Msengwa, Amina Suleiman; Vanobberghen, Fiona; Otwombe, Kennedy

VERSION 1 – REVIEW

REVIEWER	Claire Keene University of Oxford, Nuffield Department of Medicine, Centre for Tropical Medicine and Global Health
REVIEW RETURNED	24-Feb-2023

GENERAL COMMENTS	General This is a useful scoping review to publish as a reference piece. It will be really helpful to advocate for more use of longitudinal evaluation and a more nuanced approach to measuring programme delivery and success. Currently there is a lot of information but it is difficult to navigate without having to go back and forth between the methods, and this will reduce the likelihood that the findings will be taken up. The manuscript is missing a step of analysis to categorise what was found in a meaningful, helpful and implementable way (e.g. what does it mean to use 3 stages vs 6 – why does it matter?). Thinking through the usefulness of the categories and being explicit about the meaning of what you have found, will make this piece a more valuable reference. It would be acceptable to publish as is as a descriptive piece of what was scopes from the literature, but it has the potential to be much stronger The tables are well presented and add clarity to the paper. I appreciate the footnotes within the table to ensure precision and clarity. Abstract The following is unnecessary: “The 38 review findings were summarised using maps, bar graphs, and descriptive statistics.”. Rather use the words to expand on the results and conclusions. Can you add more info to the results to make it more specific and precise? For example adding numerator and denominator in brackets after percentages. Another example would be in “Of the articles that used a longitudinal design, six articles used multistate models”, could you add how many articles used a longitudinal design, e.g. or “Six of the [number] articles that used a longitudinal design, used multistate models, including...” Strengths and limitations
---

	Lines 86-87: Under limitations, it would be quite easy to assess the impact of including other languages in this review. If you conduct the search, and compare the number of titles/abstracts identified before and after you apply the English filter, you can calculate the proportion of the total number identified for screening that were not English. I suspect this will be a low number, as most of the high burden countries are Anglophone. Additionally, you only removed small numbers of 'other language' papers in your PRISMA ScR diagram. Introduction It would be good to reference the Ehrenkranz cascade, which is gaining traction. It describes the churning nature of engagement in cascade form. Ehrenkranz P, Rosen S, Boulle A, Eaton JW, Ford N, Fox MP, et al. The revolving door of HIV care: Revising the service delivery cascade to achieve the UNAIDS 95-95-95 goals. PLoS Medicine. 2021;18(5):1-10. Line 129: The references you give for reviews of measuring engagement with care and treatment (25,26) are focused on specific populations (incarcerated individuals and MSM in Africa), but are presented as general followed by other cascades in particular populations. There are other scoping studies of measuring engagement more broadly. I published the following that may be a useful reference, and dovetails well with this paper (though it looks at measuring engagement more broadly than the cascade). Please feel free to use this or ignore it, but wanted to share in case it is helpful. Keene CM, Ragunathan A, Euvrard J, English M, McKnight J, Orrell C, et al. Measuring patient engagement with HIV care in sub-Saharan Africa: a scoping study. Journal of the International AIDS Society. 2022;2022:26025-. Methods Line 157: what were your criteria for the quantitative component to be described well? Were there certain items that needed to be present? Lines 188-189: You mention the lead author reviewed all the extracted data. How many other data extractors were involved and how many/ what proportion did they each review? Line 203: do you mean that global reports had to include 170 countries or that there was a single global report that included 170 countries? Perhaps state what the minimum was to be considered global, to keep consistent with providing definitions (e.g. multiple regions = global). Also unclear why you specifically mention countries for some regions and not others – seems unnecessary. Lastly you have overlapping categories (e.g eastern and southern Africa, then sub-Saharan Africa). Perhaps separate out categories that are mutually exclusive, then have a second tier of broader regions (e.g. SSA and global). And explain whether sources could belong to more than one category (e.g. is a source from South Africa in both SSA and southern Africa?) Line 204: "The study designs included two major groups: cross-sectional and longitudinal." is more results than methods. Just define what each study design means without this line. Other queries:  - Did you not do a bibliographic review or a google search? - Can you outline the limits you put on the search, e.g. date of publication, language etc? you can do this alongside the search terms. S1: can you include the number of articles for your last row: 'other'
--	---

	search? S2: it seems from your data extraction template that you were not looking for other longitudinal measures, only multistate models. This seems like it would have introduced selection bias. Why did you not include model fit criteria like BIC/ AIC/ CAIC in your list of methods to assess goodness of fit? Were the options in the table here pre-set options or are these what you found? Figure S3: what was stage 8 in the one cascade that had 9 stages? The only names in stage 8 are viral load status – seems that the cascade with 9 stages would not have had stage 8 and 9 named the same thing? It would be better to merge figures s2 and s3, and s4 and s5 for clarity Results You have many of your own categorisations, e.g. small to large cascades, dependent vs independent staging, and different levels. With the addition of cross-sectional vs longitudinal this becomes really confusing to follow and I kept having to scroll back to the methods to remind myself what meant what. It might help to have a table of definitions instead of long paragraphs in the methods, so that the reader can use it as a quick reference, and add in more signposting/ subheadings in the results section. The more digestible you make it the easier it is to read and the more the reader will take away (and more likely they are to read it) In general, include numerator and denominators, as well as percentages, so that your writing is more precise. Figure 1: you don't need to repeat all the numbers and the reasons for exclusion from the PRISMA diagram in the text – repetitive and uses up precious words. Figure 2: the ranges are not immediately clear. I initially thought it was references to specific papers (e.g. paper 10 and 84) rather than a range. Perhaps rather present it as 10-84? And also ensure the ranges don't overlap – e.g. would 10 papers be dark green or medium green? And 3 papers be medium or light green? Lines 272-273: are ≥ 2 countries the definition of a region? Shouldn't this just be "The review included 29 studies conducted in regions or globally" Lines 276-277: it would be helpful to have the numbers of studies from each of these top countries to situate their contribution (numerator and denominator). You can save words elsewhere to include this. Lines 280-284: perhaps it would be more meaningful to present the median sample size per level of the cascade as that is likely the factor that dictates sample size? Table 1: why is the longitudinal column '-' rather than 0(0) for various items? This suggests it is not applicable rather than no studies included these categories? Same comment goes for other tables. The longitudinal column for publication year does not add up to 100%. Table 2: as not all studies were eligible for this analyses, please include the denominator (e.g. 17/x (5.9%)) so that it is clear) Lines 303-311: it might be helpful to construct an 'inclusive cascade' that includes all possible stages from the 'start' of the cascade to the end. You could then situate what it means for cascades to start with PLHIV (earlier) vs starting with diagnosis (later than having HIV but earlier than linkage). You could also categorise stages into ones that
--	--

	are sequential (like having HIV □ knowing you have HIV/diagnosis □ linking to care □ starting treatment) and those that are alternatives (being VL suppressed vs VL unsuppressed vs LTFU vs death). This would add a layer of analysis by categorising your findings against some sort of reference or template. By anchoring the categories that you use in your results, they become more meaningful. As an analogy, you have separated your puzzle pieces into different categories (like blue ones and green ones and brown ones) but haven't taken the next step of trying to make sense of them (e.g. situating them relative to each other as sky, building and grass). Discussion and conclusion Lines 362-363: these examples need more explanation – otherwise the reader either ignores them or has to look them up. From line 370: just justify a little more why you spend a large part of the discussion on multistate models and not other longitudinal methods that were found in the scoping study – was this because they proved to be particularly useful for the sources included in this study and that is a finding of your review? Or is it your personal belief that they are a good way forward? 393: same comment as above re: inclusion of denominators to make your writing more precise Conclusion: you say that a methodological guide on multi-state models and the HIV cascade is needed – but as per the above comment, you just need to justify why multi-state models are your recommendation over other longitudinal methods. This can be a simple justification, but something is needed. You presented many findings in the results section that you did not explore in the discussion – worth reviewing whether they are all necessary, and conversely if some should be discussed.
--	---

REVIEWER	Omeid Heidari University of Washington Seattle Campus, Child, Family, and Population Health
REVIEW RETURNED	03-Mar-2023

GENERAL COMMENTS	Intro: If Granich (34) considered articles that looked at cascade and continuum, should this review pick up after 2016 rather than also consider the articles in than include them in this review? The continue on the last point in the methods, what is the reason for considering the older time points if there are 1) systematic reviews that cover these and 2) this review attempts to capture latest methods, in particular care continuum with longitudinal models. “in this review, studies that included cascade assessments using both cross-sectional and longitudinal designs were included in the longitudinal study design group” Could you please expand on this? Why would a cross-sectional study be included in longitudinal? Do you mean studies that considered multiple cross sectional time points (as opposed to multiple measures from individuals) were considered longitudinal While I understand that the previous convention was “injecting drug users” the term “people who inject drugs” is less stigmatizing. Please consider using this term. Why were articles in press excluded? These were articles accepted for publication and peer reviewed were they not? I have not commonly seen this exclusion.
--

	Were there further exclusion criteria for the full text phase? Generally reasons for exclusion are included here as well, with most common reasons reported. A question about the surveillance manuscripts included. Are these that analyze data from surveillance databases but must be peer reviewed? Or did you include surveillance reports that are indexed, such as the Centers for Disease Control and Prevention’s HIV surveillance reports? Table one for Data sources has a line for Simulated data, but that was an exclusion criterion, no? Why were those 11 studies included? What percentage of longitudinal studies used time spent in each stage? For the tests in the next sentence, what were these tests generally used for? As I look through the references and tables I see that there is not a list of all the articles reviewed for this study. Suggest for the sake of transparency that either the extraction document or at least the articles and their citations included in this study be published, even as a supplement. Will defer to the editor whether they should be cited in the manuscript or in a supplement. Discussion “Researchers have proposed a framework that incorporates the in-and-out of HIV....” You should cite the researchers and framework Overall this is a well written article that captures differences in measurement and reporting of hiv care cascaded and care continuums. There are questions to the authors and suggestions to consider. Happy to review future versions.
--	---

VERSION 1 – AUTHOR RESPONSE

REVIEWER: 1

Dr. Claire Keene, University of Oxford

Comments to the Author:

General

Comment #1

This is a useful scoping review to publish as a reference piece. It will be really helpful to advocate for more use of longitudinal evaluation and a more nuanced approach to measuring programme delivery and success.

Response: Thank you very much for the positive feedback

Comment #2

Currently there is a lot of information but it is difficult to navigate without having to go back and forth between the methods, and this will reduce the likelihood that the findings will be taken up.

The manuscript is missing a step of analysis to categorise what was found in a meaningful, helpful and implementable way (e.g. what does it mean to use 3 stages vs 6 – why does it matter?). Thinking through the usefulness of the categories and being explicit about the meaning of what you have found, will make this piece a more valuable reference. It would be acceptable to publish as is as a descriptive piece of what was scopes from the literature, but it has the potential to be much stronger

Response: Thank you for the comment. This point has been addressed by including additional discussion and interpretation of the review findings (lines #393-414).

Comment #3

The tables are well presented and add clarity to the paper. I appreciate the footnotes within the table to ensure precision and clarity.

Response: Thank you. Additional footnotes have been included in all tables (from Table 1 – 4).

Abstract

Comment #4

The following is unnecessary: “The 38 review findings were summarised using maps, bar graphs, and descriptive statistics.”. Rather use the words to expand on the results and conclusions.

Response: Thank you for the observations. We have shortened the text and kept a brief sentence on the methods as required by the review guideline.

Comment #5

Can you add more info to the results to make it more specific and precise? For example adding numerator and denominator in brackets after percentages.

Response: The numerators and denominators have been added in the results section of the abstract.

Comment #6

Another example would be in “Of the articles that used a longitudinal design, six articles used multistate models”, could you add how many articles used a longitudinal design, e.g. or “Six of the [number] articles that used a longitudinal design, used multistate models, including...”

Response: Thank you for the suggestion. The total number of longitudinal studies has been included in line #55.

Comment #7

Strengths and limitations

Lines 86-87: Under limitations, it would be quite easy to assess the impact of including other languages in this review. If you conduct the search, and compare the number of titles/abstracts identified before and after you apply the English filter, you can calculate the proportion of the total number identified for screening that were not English. I suspect this will be a low number, as most of the high burden countries are Anglophone. Additionally, you only removed small numbers of ‘other language’ papers in your PRISMA ScR diagram.

Response: Thank you for the suggestion. The bibliographical database search did not contain a language filter (see Table S1), but the search queries themselves were written using the English language (i.e., all search terms included were in the English language only). The “other language” papers that were retrieved in these queries were those articles with abstracts in English and other languages, but the main text was all written in other languages. To address this comment the search terms would have to be written in other languages too (such as in French, Chinese, Spanish, Portuguese etc). The title and abstract screening would have to be proficient in these languages to establish the proportion of articles missed. Unfortunately, the review team members were all proficient in English language only and this is why we included “other languages” as a study limitation.

Introduction

Comment #8

It would be good to reference the Ehrenkranz cascade, which is gaining traction. It describes the churning nature of engagement in cascade form.

Ehrenkranz P, Rosen S, Boule A, Eaton JW, Ford N, Fox MP, et al. The revolving door of HIV care: Revising the service delivery cascade to achieve the UNAIDS 95-95-95 goals. PLoS Medicine. 2021;18(5):1-10.

Response: The proposed reference is a very good suggestion and fits well in the discussion where we discuss different types of cascade frameworks proposed in the literature after finding different ways of constructing cascades and their limitations. We have now included the proposed reference in lines #388-392.

Comment #9

Line 129: The references you give for reviews of measuring engagement with care and treatment (25,26) are focused on specific populations (incarcerated individuals and MSM in Africa), but are presented as general followed by other cascades in particular populations. There are other scoping studies of measuring engagement more broadly. I published the following that may be a useful reference, and dovetails well with this paper (though it looks at measuring engagement more broadly than the cascade). Please feel free to use this or ignore it, but wanted to share in case it is helpful. Keene CM, Ragunathan A, Euvrard J, English M, McKnight J, Orrell C, et al. Measuring patient engagement with HIV care in sub-Saharan Africa: a scoping study. Journal of the International AIDS Society. 2022;2022:26025-.

Response: Thank you for the additional reference. The suggested reference has been included in other types of reviews in line #110.

Methods

Comment #10

Line 157: what were your criteria for the quantitative component to be described well? Were there certain items that needed to be present?

Response: The mixed method articles included were the articles that assessed the cascade quantitatively and the results presented in the manuscript. The criteria used have been clarified in lines #139-140.

Comment #11

Lines 188-189: You mention the lead author reviewed all the extracted data. How many other data extractors were involved and how many/ what proportion did they each review?

Response: Thank you. Two reviewers reviewed all included articles (double review). Please see line #160. The same reviewers extracted data from the articles they reviewed.

Comment #12

Line 203: do you mean that global reports had to include 170 countries or that there was a single global report that included 170 countries? Perhaps state what the minimum was to be considered global, to keep consistent with providing definitions (e.g. multiple regions = global).

Response: This was a single report that included 170 countries that used UNAIDS data. The information is now updated in lines #184-187.

Comment #13

Also unclear why you specifically mention countries for some regions and not others – seems unnecessary.

Response: Thank you for the comments; the text has been amended accordingly in lines #184-187.

Comment #14

Lastly you have overlapping categories (e.g. eastern and southern Africa, then sub-Saharan Africa). Perhaps separate out categories that are mutually exclusive, then have a second tier of broader regions (e.g. SSA and global). And explain whether sources could belong to more than one category (e.g. is a source from South Africa in both SSA and southern Africa?)

Response: Thank you for the comment; the text has been amended accordingly in lines #184-187 and in Table 1.

Comment #15

Line 204: "The study designs included two major groups: cross-sectional and longitudinal." is more results than methods. Just define what each study design means without this line.

Response: Thank you for the suggestion, the sentence was deleted (line #183).

Comment #16

Other queries:

- Did you not do a bibliographic review or a google search?
- Can you outline the limits you put on the search, e.g. date of publication, language etc? you can do this alongside the search terms.

Response: The Google Scholar search engine and bibliographies of the papers were reviewed, from which we identified further articles for longitudinal papers that used the multistate method. These were included in the list of other strategic searches (Table S1).

The database search queries did not have a date of publication limit because we were also interested in the time of publication as findings (especially the start of these publications). The database search was done using queries written in the English language. We did not limit the language to English and this is why we got a few articles in other languages that had abstracts in both English and other languages (see also response to comment #7 above).

Comment #17

S1: can you include the number of articles for your last row: 'other' search?

Response: The number has been added in Table S1.

Comment #18

S2: it seems from your data extraction template that you were not looking for other longitudinal measures, only multistate models. This seems like it would have introduced selection bias.

Response: Thank you for the observation. The search terms included both longitudinal and multistate concepts so that all types of analyses are captured (please Table S1 and Table S4 in the supplementary materials). Studies that used multistate methods had more information that was not easily captured with the initial data extraction tool. Therefore, the additional information that was found in these articles was added to the data extraction tool as explained in lines #170-171.

Comment #19

Why did you not include model fit criteria like BIC/ AIC/ CAIC in your list of methods to assess goodness of fit? Were the options in the table here pre-set options or are these what you found?

Response: Thank you for this comment. Model fit assessment for the multistate methods listed was what was found after reviewing articles. The model fit criteria were not the focus because most articles applied models for the association. The model fit assessment for multistate methods was also omitted.

Comment #20

Figure S3: what was stage 8 in the one cascade that had 9 stages? The only names in stage 8 are viral load status – seems that the cascade with 9 stages would not have had stage 8 and 9 named the same thing? It would be better to merge figures s2 and s3, and s4 and s5 for clarity

Response: Thank you for this point. One author Hsieh et al 2015 reported two types of viral load assessment, the lab-based viral load stage and viral load status that was known by the patients (known as operational stages definition), which were presented as stages 8 and 9. We have now combined these two as one "viral load" stage. However, with the new analyses in the manuscript, framework plots were all removed.

Results

Comment #21

You have many of your own categorisations, e.g. small to large cascades, dependent vs independent staging, and different levels. With the addition of cross-sectional vs longitudinal this becomes really confusing to follow and I kept having to scroll back to the methods to remind myself what meant what. It might help to have a table of definitions instead of long paragraphs in the methods, so that the reader can use it as a quick reference, and add in more signposting/ subheadings in the results section. The more digestible you make it the easier it is to read and the more the reader will take away (and more likely they are to read it)

Response: Thank you. Since we have reached the maximum limit of the number of tables and figures in the manuscript, we have included footnotes at the end of tables for more information (Table 1-4). In the results section, one sub-section title has been included in line #347.

Comment #22

In general, include numerator and denominators, as well as percentages, so that your writing is more precise.

Response: Numbers and percentages have been added where applicable in the results section.

Comment #23

Figure 1: you don't need to repeat all the numbers and the reasons for exclusion from the PRISMA diagram in the text – repetitive and uses up precious words.

Response: Thank you. Reasons for exclusion are now removed from the text and the text has been improved (please see lines #250-258).

Comment #24

Figure 2: the ranges are not immediately clear. I initially thought it was references to specific papers (e.g. paper 10 and 84) rather than a range. Perhaps rather present it as 10-84? And also ensure the ranges don't overlap – e.g. would 10 papers be dark green or medium green? And 3 papers be medium or light green?

Response: The map range has been revised as suggested. Please see Figure 2.

Comment #25

Lines 272-273: are ≥ 2 countries the definition of a region? Shouldn't this just be "The review included 29 studies conducted in regions or globally"

Response: Thank you. The sentence has been revised in lines #263-264.

Comment #26

Lines 276-277: it would be helpful to have the numbers of studies from each of these top countries to situate their contribution (numerator and denominator). You can save words elsewhere to include this.

Response: Thank you. The referenced data is coming from a map (Figure 2), whereby the number of studies was grouped in ranges. The statement has been revised based on the data format from Figure 2 (see lines #271-274).

Comment #27

Lines 280-284: perhaps it would be more meaningful to present the median sample size per level of the cascade as that is likely the factor that dictates sample size?

Response: The requested additional results have been included in Table 2 and lines #301-302.

Comment #28

Table 1: why is the longitudinal column '-' rather than 0(0) for various items? This suggests it is not applicable rather than no studies included these categories? Same comment goes for other tables. The longitudinal column for publication year does not add up to 100%.

Response: Thank you for the suggestions. In all the tables, '0' replaced the "-" as suggested (please see Tables 1 and 3). There was a typo in the publication year for longitudinal studies for percentages of 16 articles and it has been corrected in Table 1.

Comment #29

Table 2: as not all studies were eligible for this analyses, please include the denominator (e.g. 17/x (5.9%)) so that it is clear).

Response: Thank you. All tables have been revised to include both numerators and denominators with percentages in brackets.

Comment #30

Lines 303-311: it might be helpful to construct an 'inclusive cascade' that includes all possible stages from the 'start' of the cascade to the end. You could then situate what it means for cascades to start with PLHIV (earlier) vs starting with diagnosis (later than having HIV but earlier than linkage). You could also categorise stages into ones that are sequential (like having HIV \square knowing you have HIV/diagnosis \square linking to care \square starting treatment) and those that are alternatives (being VL suppressed vs VL unsuppressed vs LTFU vs death). This would add a layer of analysis by categorising your findings against some sort of reference or template. By anchoring the categories that you use in your results, they become more meaningful. As an analogy, you have separated your puzzle pieces into different categories (like blue ones and green ones and brown ones) but haven't taken the next step of trying to make sense of them (e.g. situating them relative to each other as sky, building and grass).

Response: We thank the reviewer for this comment. Additional data items have been added in lines #218-224 and analyses have been included in Table 2 under types of cascade used and Table S4. The interpretation of findings has been added in the discussion section in lines # 393-405.

Discussion and conclusion

Comment #31

Lines 362-363: these examples need more explanation – otherwise the reader either ignores them or has to look them up.

Response: Thank you. All examples have now been explained (please see lines #381-392).

Comment #32

From line 370: just justify a little more why you spend a large part of the discussion on multistate models and not other longitudinal methods that were found in the scoping study – was this because they proved to be particularly useful for the sources included in this study and that is a finding of your review? Or is it your personal belief that they are a good way forward?

Response: Thank you. In this review, the findings about the use of multistate methods are as important as the findings of the longitudinal methods. We have now expanded on the longitudinal methods discussion. The multistate methods have additional considerations such as types of models used, different model formulations, and model assumptions, which we felt are important to highlight, but we have now reduced the text (please see lines #415-442).

Comment #33

393: same comment as above re: inclusion of denominators to make your writing more precise

Response: Thank you; the paragraph was deleted from the discussion to remain within the journal word count limit.

Comment #34

Conclusion: you say that a methodological guide on multi-state models and the HIV cascade is needed – but as per the above comment, you just need to justify why multi-state models are your recommendation over other longitudinal methods. This can be a simple justification, but something is needed.

Response: Thank you for this comment. We do not intend to state that multistate models are recommended over the other longitudinal methods, rather that we see the application of these methods growing, and each study uses them differently. There is currently a lack of guidance about how to apply these innovative methods in the HIV cascade and continuum of care. A methodological guide would therefore help researchers who want to apply these models in cascade assessments. In line with our response to comment 32 above, we have tried to clarify accordingly in the discussion. Furthermore, the guide has been expanded to cover longitudinal design methods (please see line 482).

Comment #35

You presented many findings in the results section that you did not explore in the discussion – worth reviewing whether they are all necessary, and conversely if some should be discussed.

Response: We have reviewed the discussion and (a) reduced the discussion on multistate methods and (b) added more discussion on other key review findings.

REVIEWER: 2

Dr. Omeid Heidari, University of Washington Seattle Campus

Comments to the Author:

Comment #1

Intro: If Granich (34) considered articles that looked at cascade and continuum, should this review pick up after 2016 rather than also consider the articles in than include them in this review?

Response: Thank you for the observation. Granich et al 2017 mainly focused on synthesizing the estimates for 90-90-90 progress. Our review covered statistical methods in broader scope which included descriptive and regression models. Therefore, we purposely wanted to include articles that may have been covered by Granich et al 2017 and therefore did not restrict our database search. We have improved the method description in lines #112-113.

Comment #2

The continue on the last point in the methods, what is the reason for considering the older time points if there are 1) systematic reviews that cover these and 2) this review attempts to capture latest methods, in particular care continuum with longitudinal models.

Response: Thank you very much for the observation. As above, our review question was broader than the others that were published before. We were not aiming to show recent methods used but to identify methods used to assess the HIV care cascade and continuum of care over the years. A finding of our review was that there have been recent developments and applications of longitudinal methods. Most of the methods were implemented after the HIV cascade researchers observed the limitations of the standard HIV care cascade framework proposed earlier (Ref: Powers KA, Miller WC. Building on the HIV cascade: a complementary “HIV States and Transitions” framework for describing HIV diagnosis, care, and treatment at the population level. *J Acquir Immune Defic Syndr*. 2015 Jul 1;69(3):341–7.).

Comment #3

“in this review, studies that included cascade assessments using both cross-sectional and longitudinal designs were included in the longitudinal study design group” Could you please expand on this? Why would a cross-sectional study be included in longitudinal? Do you mean studies that considered

multiple cross sectional time points (as opposed to multiple measures from individuals) were considered longitudinal.

Response: Yes, more explanation has been included in lines #188-189 as suggested. There were few studies (n=3) that conducted both cross-sectional and longitudinal analyses. It is known that once you have longitudinal data it is very easy to derive cross-sectional cascade results at a selected time of interest. The authors of the articles mainly focused on longitudinal analysis and added the cross-sectional analyses at the latest time point to show how the two differ and complement each other, especially for repeated cross-sectional analyses over the years. Since the longitudinal part was the major part of the articles and the cross-sectional design was within the longitudinal design, we decided to include these papers under the longitudinal design.

Comment #4

While I understand that the previous convention was “injecting drug users” the term “people who inject drugs” is less stigmatizing. Please consider using this term.

Response: Thank you for the suggestion, It is now corrected in lines #198, 260, and in Table 1.

Comment #5

Why were articles in press excluded? These were articles accepted for publication and peer reviewed were they not? I have not commonly seen this exclusion.

Response: Thank you for your comment. Only one database (EMBASE) retrieved this type of article. It was not clear to the team which submission stage these papers were in (for example, if they were still under review). Which could jeopardize the accessibility to full papers for review. Therefore, we did not include them.

Comment #6

Were there further exclusion criteria for the full text phase? Generally reasons for exclusion are included here as well, with most common reasons reported.

Response: Thank you. The exclusion criteria applied at each stage were the same. As per Figure 1, 10 articles were excluded at the stage of full-text reviews with the reasons shown in Figure 1.

Comment #7

A question about the surveillance manuscripts included. Are these that analyze data from surveillance databases but must be peer reviewed? Or did you include surveillance reports that are indexed, such as the Centers for Disease Control and Prevention’s HIV surveillance reports?

Response: Thank you for the comment. As per the methods, we only included published articles, although some published articles described their data sources as being surveillance data. We have included this as a limitation in the discussion.

Comment #8

Table one for Data sources has a line for Simulated data, but that was an exclusion criterion, no? Why were those 11 studies included?

Response: Thank you for the observation. We realized that there was a mistake in including 4 papers that were modeling studies and which have now been excluded, and the PRISMA flow diagram updated. The remaining 7 papers were misplaced in modeling data sources due to a modeling method described in articles that were used for estimating the number of PLHIV or undiagnosed PLHIV (stage one of the HIV care cascade). This has now been corrected.

Comment #9

What percentage of longitudinal studies used time spent in each stage?

Response: Thank you for the question. 24% of longitudinal studies used the time spent in each stage; the text has been updated accordingly (line #334).

Comment #10

For the tests in the next sentence, what were these tests generally used for?

Response: In articles that used simple test statistics, the tests were used to compare cascade stages in terms of characteristics.

Comment #11

As I look through the references and tables I see that there is not a list of all the articles reviewed for this study. Suggest for the sake of transparency that either the extraction document or at least the articles and their citations included in this study be published, even as a supplement. Will defer to the editor whether they should be cited in the manuscript or in a supplement.

Response: Thank you for the suggestion. The full list of all articles included in this review and their full citation is now included in the supplementary materials in Table S3. The text has been updated by citing the supplementary table in line #258. Due to this review, 9 articles were identified as duplicates and they have been removed from the list.

Discussion

Comment #12

“Researchers have proposed a framework that incorporates the in-and-out of HIV...” You should cite the researchers and framework.

Response: Thank you for your comment. The papers have been cited as suggested (please see line #386).

Comment #13

Overall this is a well written article that captures differences in measurement and reporting of hiv care cascaded and care continuums. There are questions to the authors and suggestions to consider. Happy to review future versions.

Response: Thank you very much for the positive comment.

VERSION 2 – REVIEW

REVIEWER	Omeid Heidari University of Washington Seattle Campus, Child, Family, and Population Health
REVIEW RETURNED	12-Jul-2023
GENERAL COMMENTS	Thank you for incorporating revisions. Regarding point #5 from my comments, in press articles are accepted articles that have not been published yet but have been peer reviewed and are available on line. I would suggest a final limitation regarding this since this is the latest available evidence that has been excluded from screening.